# A Systematic Review of the Distribution of Tick-Borne Pathogens in Wild Animals and Their Ticks in the Mediterranean Rim between 2000 and 2021

**DOI:** 10.3390/microorganisms10091858

**Published:** 2022-09-16

**Authors:** Baptiste Defaye, Sara Moutailler, Vanina Pasqualini, Yann Quilichini

**Affiliations:** 1UMR CNRS Science Pour l’Environnement 6134, Université de Corse Pascal Paoli, Projet GEM, 20250 Corte, France; 2ANSES, INRAE, Ecole Nationale Vétérinaire d’Alfort, UMR BIPAR, Laboratoire de Santé Animale, 94700 Maisons-Alfort, France

**Keywords:** island, Mediterranean Rim, TBP, tick, wild animal

## Abstract

Tick-borne pathogens (TBPs) can be divided into three groups: bacteria, parasites, and viruses. They are transmitted by a wide range of tick species and cause a variety of human, animal, and zoonotic diseases. A total of 148 publications were found on tick-borne pathogens in wild animals, reporting on 85 species of pathogens from 35 tick species and 17 wild animal hosts between 2000 and February 2021. The main TBPs reported were of bacterial origin, including *Anaplasma* spp. and *Rickettsia* spp. A total of 72.2% of the TBPs came from infected ticks collected from wild animals. The main tick genus positive for TBPs was *Ixodes*. This genus was mainly reported in Western Europe, which was the focus of most of the publications (66.9%). It was followed by the *Hyalomma* genus, which was mainly reported in other areas of the Mediterranean Rim. These TBPs and TBP-positive tick genera were reported to have come from a total of 17 wild animal hosts. The main hosts reported were game mammals such as red deer and wild boars, but small vertebrates such as birds and rodents were also found to be infected. Of the 148 publications, 12.8% investigated publications on Mediterranean islands, and 36.8% of all the TBPs were reported in seven tick genera and 11 wild animal hosts there. The main TBP-positive wild animals and tick genera reported on these islands were birds and *Hyalomma* spp. Despite the small percentage of publications focusing on ticks, they reveal the importance of islands when monitoring TBPs in wild animals. This is especially true for wild birds, which may disseminate their ticks and TBPs along their migration path.

## 1. Introduction

Pathogens are one of the biggest worldwide heath threats. Zoonotic pathogens target both humans and animals, whereas non-zoonotic pathogens infect either animals or humans. Zoonotic pathogens represent more than half of the pathogens worldwide and provoke mainly emerging diseases [1]. Some bacteria, parasites, and viruses can circulate by the way of arthropod vectors with hematophagous behavior [2], such as mosquitoes, ticks, or sandflies due to their blood-sucking practices. Vector-borne diseases annually cause over 17% of infections and 700,000 deaths across the globe [3]. Tick-borne pathogens (TBPs) are not only of veterinary importance—they also include zoonotic pathogens harmful for humans [4].

Ticks (Ixodida) are the main carrier of veterinary pathogens and the second most important vector of human pathogens, right after mosquitoes [5,6]. These hematophagous arthropods belong to the Ixodida order, which is composed of three families. The first, Ixodidae, is further divided into the Prostriasta composed of the subfamily Ixodinae and the genus Ixodes; and the Metastriata composed of all other subfamilies and eight genera, *Anamalohimalaya*, *Cosmiomma*, *Dermacentor*, *Hyalomma*, *Margaropus*, *Nosomma*, *Rhipicentor*, and *Rhipicephalus*. These “hard ticks” represent most tick species. The second family, Argasidae, is divided into two subfamilies: the Argasinae, including the genus *Argas*, and Ornithodorinae, composed of the genera *Ornithodoros*, *Otobius*, and *Carios*. These are known as “soft ticks”. The third family is the Nuttallielliedae family, composed of one species [7]. The tick’s life cycle has four stages: egg, larva, nymph, and adult male or female. Their tropisms vary according to the species and life cycle stage. Ticks can have different tropisms to one, two, or more hosts. They can transmit various bacteria, viruses, and parasites [5] to a broad range of wild animals (whether mammalian, reptilian, amphibian, or avifaunal) [4,8,9,10,11].

Ticks and tick-borne diseases can spread by way of migratory birds. Their long-distance migration allows migratory birds to carry infected immature argasid and ixodid ticks to Europe from Africa or vice versa [12]. Migratory birds can be one of the ways ticks and tick-borne pathogens are disseminated because of the seasonality of their migration and the presence of stopover places (mainly wetlands) [12].

According to its geographic situation between diverse continents (Europe, Africa, and Western Asia), the Mediterranean Sea is enclosed by a large number of countries with a large spectrum of biotopes, ranging from a Mediterranean to an arid climate. The Mediterranean Rim is one of the areas most impacted by climate change, human activity, and animal migration [13,14]. While the increase in temperatures in Northern Europe fosters the development of ticks such as the genus *Ixodes*, further south, in the Mediterranean Rim, the development of a dry area with plants suited to an arid climate fosters the development of ticks such as the genus *Hyalomma* [15]. The fact that immature stages can feed on birds and remain attached to their host for a long period supports the idea of ticks’ dissemination by migratory birds [16].

This review focuses on the Mediterranean Rim, where various TBPs have been detected from wild animals and their ticks. To review the geographic distribution of these TBPs and highlight possible changes in their distribution in the future, four areas were taken under consideration: the Balkans, composed of Albania, Bosnia-Herzegovina, Croatia, Greece, Montenegro, and Slovenia; the Middle East, composed of Cyprus, Israel, Lebanon, Palestine, Syria, and Turkey; North Africa, composed of Algeria, Egypt, Libya, Morocco, and Tunisia; and finally, Western Europe, composed of France, Italy, Malta, Monaco, and Spain. The different areas were determined by geographic proximity and common biotope. The last part of our review entailed focusing on the monitoring of TBPs in wild animals and their ticks on the western and eastern Mediterranean islands. This was particularly important for avifauna, as the islands could become hotspots during bird migrations.

The purpose of this study was to review the literature between 2000 and early 2021 on the presence of TBPs in wild animals and their ticks in the Mediterranean Rim, following PRISMA guidelines, with the objectives listed below:-Perform a bibliometric analysis of TBP studies.-Review the diversity of TBPs from wild animals and their ticks (engorged tick species positive for pathogens, and wild animal hosts with TBPs in the Mediterranean Rim).-Compare the distribution of TBPs found in wild animals and their ticks in the four main areas defined.-Focus on the distribution of TBPs on the Mediterranean islands.

## 2. Materials and Methods

We performed a literature review concerning TBPs in all the countries of the Mediterranean Rim (n = 22). We followed PRISMA guidelines [17] for the identification and selection of studies relevant to the topic. We compiled and analyzed the data from the studies included in this review.

All the articles published in international journals indexed by Scopus, PubMed, and/or Web of Science were taken into consideration (Figure 1). Only English articles were taken into account. The research date range was from 1 January 2000 to 31 February 2021. The keywords used were Pathogens AND Ticks AND each Mediterranean Rim country, with the “all fields” option to select the articles in which search items appeared in the title, abstract, and keywords. All papers considered case studies, literature review, “gray” literature, guides from relevant organizations, and abstracts in poster form from conferences were excluded. We also discarded diagnosis in humans and animals and clinical descriptions of disease. We selected studies targeting the distribution of TBPs and TBP-positive ticks in the Mediterranean Rim and Mediterranean islands. We excluded all duplicates and inaccessible articles (due to the languages or unavailability of the full text). Titles and abstracts were reviewed, and we applied inclusion/exclusion criteria to the 1070 articles selected. Filtering was based on the answers to the following questions:Did the study include a country along the Mediterranean coastline? Yes/NoDid the study include tick-borne pathogens? Yes/NoDid the study exclude ticks collected among vegetation (flagging)? Yes/No

Publications targeting TBPs in wild animals and/or in engorged ticks collected from wildlife in Mediterranean countries were included. This gave us an overview of research on TBP-positive ticks supposedly infesting hosts and TBPs circulating among the wild animal population. The articles were saved if the answer to all three questions was “yes” and eliminated otherwise. The next step entailed reviewing the information of interest from the full text of 299 articles into a database. We also excluded articles that did not fit the criteria.

We next reviewed the bibliography of each selected article in order to check for new relevant articles to include in the review. The same steps previously described were used for these new articles. For the last step, we excluded articles on domestic animals (n = 232) and selected only articles dealing with wild animals (n = 148).

All the different selection steps are summarized in Figure 1, with an explanation of inclusion/exclusion criteria applied to articles for this review.

The data of interest were compiled in an Excel table that had previously been tested using 15 articles, and included the following information:Main characteristics of the studies: article ID, years, authors, analytical, and statistical methodology.Pathogen-related information: pathogens screened and detected, species, number of species, zoonotic status, host.Tick-related information: species, type, number, stage.Host-related information: groups, sedentary or migratory.Area of interest: country, type of area, number of sampling sites.

The different outputs of the data worksheet were selected following mutual agreement from all the authors.

## 3. Bibliographic Analysis

The first article considered, published in 2001, was the only paper selected for that year. During the first decade investigated, there were few papers between 2001 and 2005, with a peak of three papers in 2004. The number of publications began to increase between 2006 and 2013, with a mean of 7.1 per year (ranging from five to eight papers per year). Of all the papers about TBPs in wild animals from the Mediterranean Rim, 57.4% were published between 2014 and 2020. The rate of papers published doubled from eight per year in 2013 to 16 per year in 2014. Between 2016 and 2020, the number of papers published per year continued to increase until it reached the highest value of both decades, with 18 publications in 2019. No paper was found in the first two months of 2021 (Figure 2).

This trend can be explained by three reasons: the increase in scientific interest about TBPs among wild animals and their ticks; the increase in accessibility to scientific journals and full-text publications; the evolution in TBP detection methodologies over the years and the development of molecular biology methodology.

Indeed, TBPs may be detected with three different types of methodology. Serological analysis is used to detect pathogens in fluids, tissue, and ticks by the way of antibodies, e.g., ELISA and immunofluorescence. Microscopic analysis was rarely used in the papers selected, although it is the most accurate method to confirm the presence of TBPs in samples. Molecular analysis covers techniques that detect the nuclear acids of TBPs in both ticks and animals, e.g., next-generation sequencing (NGS), polymerase chain reaction (PCR), and high-throughput sequencing. These molecular techniques were the most frequently used in the selected publications, even though various techniques are available to monitor TBPs. The positive evolution in publications through the years appears to be due to the growing popularity of the molecular approach (Figure 2).

## 4. Tick-Borne Pathogens from Wild Animals and Their Ticks in Mediterranean Rim Countries

A total of 148 publications were analyzed. Among them, 66.7% were about bacteria, 21% about parasites, and 12.3% about viruses. Eighty-five pathogens were reported in the Mediterranean Rim, with nine genera of bacteria, eight of parasites, and four of viruses (Table 1). Of these, 51.1% are zoonotic and 48.9% are of non-zoonotic/unknown status (Table 1). The percentage of publications about pathogens detected exclusively in ticks was 38.8%, which was higher than those concerning TBPs detected exclusively in either hosts (27.1%) or in both hosts and ticks (34.1%).

### 4.1. Parasites



**Nematoda**





**
*Cercopithifilaria*
**



Parasites of the genus *Cercopithifilaria* from the Onchocercidae are tick-borne filarioids that have mainly been described in dogs, with two frequently reported species: *Cercopithifilaria bainae* and *Cercopithifilaria grassi* [18]. In the wild animals of the Mediterranean Rim, however, only one species was reported: *Cercopithifilaria rugosicauda*, a roe deer parasite transmitted by *Ixodes ricinus* ticks. It was found in the subcutaneous samples from a dead roe deer in Italy [19]. It was only found in this one animal and never detected in ticks collected from wild animals (Table 1). The sparse information on the presence of these parasites can be explained by their low interest and impact on wildlife health. This genus is one of the three least reported in the complete dataset (0.7%) and in the dataset concerning parasites (1%).



**Apicomplexa**





**
*Babesia*
**



*Babesia* is a genus composed of protozoal parasites from the Babesiidae family transmitted by ticks. The zoonotic species are principally transmitted by ticks belonging to the family Ixodidae. The *Babesia* genus is composed of a number of species detected in wild animals and their ticks in the Mediterranean Rim, but each species was only found in between one and four countries. Among the 13 species found, *Babesia microti*—a zoonotic species responsible for babesiosis in both humans and animals—was detected in the largest range of countries: Croatia, Italy, Spain, and Turkey. The highest prevalence in ticks was found in *Ixodes hexagonus* from red foxes in Spain (55.6%), and among wild animals, from red foxes in Italy (54%) [20,21]. The other *Babesia* species were found in a large panel of hosts including birds, wild ungulates, and rodents; they were mainly found in ticks of the genus *Ixodes*. *Babesia* was the parasite genus found the most frequently; it was reported at a level of 10.8% in the complete dataset and 51.6% in the dataset concerning parasites.



**
*Cytauxzoon*
**



Parasites from the genus *Cytauxzoon* are piroplasmids from the family Theileriidae infecting wild and domestic felines, and their ticks are found in both the New and the Old Worlds [145]. In wild animals in the Mediterranean Rim, this genus was only found in lynxes from Spain, with a prevalence of 15%. These pathogens were only identified at the genus level and did not appear to be of particular interest to scientists. It is one of the three genera reported the least in the complete dataset (0.7%) and in the dataset concerning parasites (1%).



**
*Hemolivia*
**



This genus belongs to the family Karyolysidae and is composed of haemogragines, which are parasites with a tropism for reptiles as intermediate hosts and hard ticks as final hosts [146]. In wild animals of the Mediterranean Rim, only one species was found: *Hemolivia mauritanica*. This species is closely related to tortoises, and especially to the genus *Tetsudo*, where it was found with a prevalence of 17.2% in Palestine (Paperna, 2006). The scarce research on and detection of this genus are probably related to the specific tropism of this genus for reptiles. It is the least reported in the complete dataset (0.7%) and in the dataset concerning parasites (1%).



**
*Hepatozoon*
**



The genus Hepatozoon is from the Hepatozoidae family and closely related to the genus Hemolivia [146]. However, it mainly focuses on wild and domestic carnivores. Only two species from wild animals were found in the publications concerning the Mediterranean Rim: Hepatozoon canis, found in foxes in three countries, with the highest prevalence (49%) in Italy, and Hepatozoon kisrae, found in lizards, with a prevalence of 33.3% in Palestine [38,40]. Like the previous genus, Hepatozoon pathogens were found mainly in animal samples. This genus was poorly reported, being mentioned in 2.7% of the complete dataset and 4% of the dataset concerning parasites.



**
*Theileria*
**



Like *Babesia*, the genus *Theileria* from the family Theileriidae belongs to the piroplasmids group [147]. These parasites have an obligatory period in ticks and infect mammals. They provoke theileriosis (benign to fatal) in breeding animals [148]. Six *Theileria* species were detected in wild animals. Three species of the six were only found in ticks, one species only in wild animals, and two species in both. *Theileria annae* was the most frequently detected. Its highest prevalence values were 9.1% in *Ixodes ricinus* from rodents in Italy and 14% in foxes from [44,45]. This genus was the second most reported parasite genus (8.8%) in the complete dataset and was reported in 13.1% of the dataset concerning parasites.

### 4.2. Bacteria



**
*Anaplasma*
**



The genus *Anaplasma* is composed of intracellular bacteria from the family Anaplasmataceae, order Rickettsiales. These bacteria (e.g., *Anaplasma marginale*, *Anaplasma bovis*, and *Anaplasma ovis*) have a high impact in veterinary healthcare but also in human healthcare (e.g., *Anaplasma phagocytophilum* and *Anaplasma platys*) [149,150]. The most frequently reported species in wild animals was *Anaplasma phagocytophilum*, the causative agent of human granulocytic anaplasmosis. This species was detected in six countries (France, Greece, Israel, Italy, Slovenia, and Spain) of the twenty-two in the Mediterranean Rim. Eight tick species were reported to be positive for this pathogen, the highest prevalence being in *Ixodes ricinus* (11%) collected from red and roe deer in Spain [32]. Throughout the Mediterranean Rim, the main hosts of ticks positive for *A. phagocytophilum* were wild ungulates (Table 1). In wild animals, *A. phagocytophilum* was mainly found in rodents, with a prevalence of 22.8% in France. The second most widely distributed species was *Anaplasma ovis*, found in two countries (Cyprus and Italy), and responsible not only for ovine anaplasmosis in goats and sheep but also in wild ungulates. The highest prevalence was found in *Haemaphysalis sulcata*, a tick collected from mouflons in Cyprus (24%), and in *Ixodes festai* collected from hedgehogs in Italy (33.3%). In wild animals, it was only found in mouflons from Cyprus with a prevalence of 10% [70,71]. The genus *Anaplasma* is widely distributed across the Mediterranean Rim and well researched. It was reported in 19.6% of the complete dataset and 29.3% of the dataset concerning bacteria. It was the second most frequently reported bacteria genus. However, most research focused on the zoonotic species (*Anaplasma phagocytophilum*) rather than animal-specific species. Nevertheless, an equal amount of research focused on ticks and wild animals.



**
*Bartonella*
**



*Bartonella* are bacteria belonging to the Bartonellaceae family, Rhizobiales order. Nearly half of them have a zoonotic behavior [151]. They can provoke diseases such as trench fever and cat-scratch fever. In the Mediterranean Rim, a total of eight species in this genus were found in three countries: Algeria, Israel, and Italy. The species found only in Algeria was *Bartonella tamiae*, a zoonotic species primarily found in humans in Thailand. It was reported to have been found in *Ixodes vespertilionis* ticks collected from bats, with a prevalence of 63.2% [75,152]. The six species detected in Italy were all from ticks (Table 1). The highest prevalence in ticks was for *Bartonella bovis* from the species *Dermacentor marginatus*, *Haemaphysalis punctata*, and *Ixodes ricinus* collected from fallow deer and red deer [59]. The last species detected was *Bartonella elizabethae*, which was the only one reported in wild animals, with a prevalence of 25% in rodents in Israel [74]. The research on this genus mainly focused on ticks and involved only a few countries. It was reported in 4.1% of the complete dataset and in 6.1% of the dataset concerning bacteria.



**
*Borrelia*
**



The genus *Borrelia* includes spirochetes bacteria belonging to the family Spirochaetaceae. It is separated into two groups: the Lyme borreliosis group composed of bacteria from the *Borrelia burgdoferi sensus lato* group, and the relapsing fever group that includes *Borrelia miyamotoi* [153]. A total of 12 Borrelia species were found in the Mediterranean Rim. Most of them belonged to the *Borrelia burgdoferi* s.l. group, which is the most widely distributed and responsible for Lyme disease (Table 1). Among these species, *Borrelia afzelii* was the most frequently reported in four countries: France, Italy, Spain, and Turkey. It was only detected in *Ixodes ricinus*, with the highest prevalence of 4.8% in ticks collected from birds in Italy. In wild animals, *Bo. afzelii* was only detected in rodents, with the highest prevalence of 4.5% being found in Spain [11,76]. Among the other Borrelia species, the one with the highest reported prevalence in *Ornithodoros erraticus* s.l. was *Borrelia crocidurae* from Morocco (41%) and *Borrelia merionesi* in rodents from Morocco (4%) [77]. In the Mediterranean Rim, research focused on the *Borrelia* species of the Lyme disease group, which were found in both ticks and wild animals. This focus is probably linked to their importance for human health. The Borrelia genus was reported in 18.2% of the complete dataset and 27.2% of the dataset concerning bacteria. It was the third most reported bacteria genus.



**
*Chlamydia*
**



The genus *Chlamydia* belong to Chlamydiales, which are bacteria responsible for various diseases in virtually the whole animal realm (Chisu et al., 2018b). Rare studies showed that ticks could be vectors of *Chlamydia* [154]. Two species were found: *Chlamydia abortus* and *Chlamydia psitacci*. The first one—responsible for abortive chlamydiosis in small ruminants—was found only in ticks, the highest prevalence being 21.4% in *Haemaphysalis sulcata* from hedgehogs in Italy [93,155]. The second one, responsible for avian psittacosis and human pneumonia, was found in animals; the highest prevalence was 11.9% in pigeons from Italy [72,156]. These genera were only found in Italy in both wild animals and ticks, which appears to indicate that ticks may play a role in their circulation. It was reported in only 1.4% of the complete dataset and 2% of the dataset concerning bacteria.



**
*Coxiella*
**



This genus belongs to the Coxiellaceae family. Only *Coxiella burnetii*, responsible for Q fever, is representative of this genus. This TBP infects both animals and humans. The disease can provoke numerous symptoms in humans, unlike in animals, where it is mainly asymptomatic with sporadic cases of abortion and has a worldwide distribution [157]. *Coxiella burnetii* was detected in the wildlife of five countries. The highest prevalence in ticks and wild animals was observed in *Ixodes ricinus* collected from rodents in Italy and in red deer from Spain, with prevalence of 17.3% and 48.9%, respectively [86,96] (Table 1). It was reported in a large panel of countries (Algeria, Italy, Italy, Slovenia, and Spain), ticks, and hosts. This shows the importance of *C. burnetii* in the Mediterranean Rim. It was reported in 9.5% of the complete dataset and in 14.1% of the dataset concerning bacteria.



**
*Ehrlichia*
**



*Ehrlichia* is a genus belonging to the Anaplasmataceae as *Anaplasma*. It is responsible for human disease such as monocytotropic ehrlichiosis (*Ehrlichia chaffeensis* and *Ehrlichia canis*) and animal disease such as canine ehrlichiosis (*Ehrlichia canis*) [158]. The latter was the only species found in wild animals in the Mediterranean Rim (Israel and Italy). The highest prevalence in ticks was found in *Dermacentor marginatus* collected from wild boars in Italy (7.8%). The highest prevalence in wild animals was in foxes again from Italy (52%) [100,101] (Table 1). This genus was reported in 4.7% of the complete dataset and 7.1% of the dataset concerning bacteria.



**
*Neoehrlichia*
**



As *Anaplasma* and *Ehrlichia*, this genus belongs to the family Anaplasmataceae. Of the four bacteria composing this genus, only one was reported in our review: *Noehrlichia mikurensis*, a zoonotic pathogen that mainly causes chronic lymphocytic leukemia. It is mainly transmitted by the *Ixodes* genus, and rodents are well-known hosts [159]. It was found in two countries: France, with a prevalence of 1.8% in rodents, and Italy, with a prevalence of 5.3% in *Ixodes ricinus* collected from rodents [32,104] (Table 1). It was only reported in 1.4% of the complete dataset and 2% of the dataset concerning bacteria.



**
*Leptospira*
**



The genus *Leptospira* represents zoonotic spirochetes responsible for leptospirosis all over the world from the Leptospiraceae family. It is mainly transmitted by contact with a contaminated environment, but can also be detected in ticks [160,161]. A total of four bacteria were reported in our review. Two of them were reported only in Croatia among rodents. *Leptospira interrogans* and *L. borpetersenii* were reported in Croatia and France, with the highest prevalence of 30% for *Leptospira interrogans* in rodents from Croatia [29] (Table 1). Like the previous genus, it was reported in 1.4% of the complete dataset and 2% of the dataset concerning bacteria.



**
*Rickettsia*
**



The *Rickettsia* genus from the family Rickettsiaceae is one of the most important TBPs from an economic and health point of view. It includes many species separated into different groups: the first one is the spotted fever group (SFG), responsible for Mediterranean spotted fever, and the typhus group (TG), which is less well known. [162,163]. Of the 13 *Rickettsia* species or *Candidatus* species reported in our review, *Rickettsia africae* and *Rickettsia massiliae* were reported in a total of five countries each: Algeria, Greece, Israel, Italy, and Turkey for *R. africae*, and Algeria, Cyprus, Israel, Italy, and Spain for *R. massiliae*. The first one, responsible for African tick-bite fever, was found with the highest prevalence of 25% in *Hyalomma detritum* collected from wild boars in Israel [117]. *Rickettsia massiliae* is one of the rickettsia responsible for Mediterranean spotted fever and it was frequently detected in ticks from the *Rhipicephalus sanguineus* s.l. group on wild boars in Algeria (88.9%) [75]. However, none of these rickettsiae were found in animal samples. The only species found in wild animals was *Rickettsia slovaca*, an agent of tick-borne lymphadenopathy (TIBOLA), with a prevalence of 5.4% in wild boars from Algeria and 6% in rodents from Italy [83,129]. This genus was the most frequently detected in the Mediterranean Rim with the highest diversity of species during our research period, and reported most in both the complete dataset (30.4%) and the dataset concerning bacteria (45.5%). This may reflect scientists’ particular concern about this genus. Nevertheless, the genus *Rickettsia* was reported more often in ticks than in wild animals.

### 4.3. Viruses



**Flavivirus**



Flaviviruses constitute a major part of arboviruses. They are transmitted by various hematophagous vectors. They mainly target mammals and are responsible for many past outbreaks [164]. Three viruses in this genus were detected in wild animals and their ticks in the Mediterranean Rim: tick-borne encephalitis virus (TBEV), Louping ill virus, and the Maeban virus. The first two are well-known tick-borne viruses responsible for many encephalitis cases in the world. TBEV was detected in wildlife in France and Slovenia, with the highest prevalence of 5.9% in rodents from Slovenia [136]. Like TBEV, the Louping ill virus was only detected in wild animals and only in France, with a prevalence of 0.9% in wild boars and 0.1% in roe deer [132]. Little information is available on the Maeban virus. However, it was detected in seagull eggs in four countries: Algeria, France, Italy, and Spain. The highest prevalence was observed in Spanish seagull eggs (17.5%). In ticks, it was found only in *Ornithodoros maritimus* again in Spain, with the highest prevalence of 5.9% [133]. The flavivirus genus was the least reported, representing 4.7% of the complete dataset and 38.9% of the dataset concerning viruses.



**Orthonairovirus**



The Orthonairovirus contains a wide range of arboviruses transmit by ticks. One of the most well-known is the Crimean-Congo hemorrhagic fever virus (CCHFV), transmitted mainly by the genus *Hyalomma*. It provokes severe to fatal human disease throughout most of the Old World and its range has been modified with the circulation of the *Hyalomma* genus due to the climate change [165]. In wild animals living in the Mediterranean Rim, it was reported in five different countries: Algeria, Italy, Morocco, Spain, and Turkey. In animals, the highest prevalence reported was in tortoises from Turkey (9.5%) and the *Hyalomma aegyptium* tick found on tortoises in Algeria (28.6%) [140,141]. CCHFV was the most reported virus in 5.4% of the complete dataset and 61.1% of the dataset concerning viruses.

## 5. Ticks Involved in Tick-Borne Diseases among Wild Animals Living in Mediterranean Rim Countries

All the previous pathogens could be transmitted by the 32 species of ticks from eight genera reported to be found on wild animals from the Mediterranean Rim in this review. However, it is important to remember that ticks can also contain endosymbiotic intracellular bacteria, which are harmless and potentially needed for tick survival [166]. In past research, they could have been mistaken for harmful pathogens to humans and wild animals. Furthermore, due to the ticks’ feeding habits, wild animals’ interaction with other groups, and the open environment where they live, it is nearly impossible to confirm that the engorged TBP-positive ticks actually fed and became positive on the wild animals they were collected from.

From the eight tick genera, six belonged to the hard tick group (Ixodidae): *Amblyomma*, *Dermacentor*, *Haemaphysalis*, *Hyalomma*, *Ixodes*, and *Rhipicephalus*. They were widespread in the Mediterranean Rim. Soft ticks (Argasidae) were represented by the last two genera collected in Algeria and Egypt: *Argas* and *Ornithodoros* (Appendix A). In this section, the TBP-positive engorged tick species reported in the highest number of publications in a considered country was defined as the country’s main tick species.

### 5.1. Ixodidae



**
*Ixodes*
**



*Ixodes* was the most important genus collected from wild animals in terms of TBPs reported (Appendix A). This genus is well-known for its transmission of bacteria responsible for Lyme disease [167]. The ticks are mainly present in the northern part of the Mediterranean Rim and have a ubiquitous tropism with a high impact on both human and animal health [168,169]. In our data, they were reported in eight countries (Cyprus, France, Greece, Italy, Slovenia, Spain, and Turkey). Eight species were detected in the Mediterranean Rim: *I. acuminatus*, *I. festai*, *I. frontalis*, *I. hexagonus*, *I. ricinus*, *I. simplex*, *I. ventalloi*, and *I. vespertilionis* on a total of 23 hosts. Among the TBPs detected in ticks, 63.9% were detected in these eight tick species. The main species found was *I. ricinus*, in which 32 TBPs were reported of the 39 reported in the genus (Appendix A). It was also the main tick species reported in Italy, France, Slovenia, and Spain in our review (Figure 3). This ubiquitous species is mainly known for its transmission of *Borrelia burgdoferi* s.l., the causative agents of Lyme disease [167]. However, it can be reported positive for numerous pathogens detected in wild animals. In addition to *I. ricinus*, two other *Ixodes* species were defined as main tick species: *I. frontalis* in Greece and *I. ventalloi* in Cyprus, despite the few TBPs reported in these species (Appendix A, Figure 3). It shows the importance of *I. ricinus* in the monitoring of TBPs, especially in European countries.



**
*Rhipicephalus*
**



The second-ranking genus in terms of pathogen diversity reported to be found on wild animals from Mediterranean Rim countries was *Rhipicephalus* (Appendix A). It is mainly present in Africa and Europe, although *Rhipicephalus sanguineus* s.l. is also present in America and Oceania [170,171]. Like the genus *Ixodes*, this genus transmits diseases with a global health impact, including anaplasmosis, babesiosis, ehrlichiosis, and rickettsiosis [168,169]. A total of six species—*Rh (B). annulatus*, *Rh. bursa*, *Rh (B). kohlsi*, *Rh. ovis*, *Rh. pusillus*, and *Rh. sanguineus* s.l.—were reported in seven countries (Cyprus, Greece, Israel, Italy, Palestine, Spain, and Turkey). It has been defined as the main tick species in Algeria and Israel (Figure 3). This genus accounted for 36.1% (22 species) of the reported TBPs in ticks. Among them, 18 were detected in *Rh. sanguineus* s.l. (Appendix A). This is a tick species known for its transmission of *Ehrlichia canis*, *Rickettsia conorii*, and *Babesia canis*. It is usually found among domestic animals [168]. Among the other *Rhipicephalus* species, 14 TBPs were detected in *Rh. bursa*, which was one of the main tick species in Algeria (Figure 3). This shows the importance of these two species in the monitoring of *Rhipicephalus*.



**
*Haemaphysalis*
**



*Haemaphysalis* genus members are small ticks that target a wide range of hosts during their immature stage and only mammals during their adult stage [169]. Five species were detected on wild animals from the Mediterranean Rim: *H. adleri*, *H. erinacei*, *H. parva H. punctata*, and *H. sulcata* in seven countries (Cyprus, Greece, Israel, Italy, Slovenia, Spain, and Turkey). They were collected from 13 wild animals with a total of 34.4% of the TBPs reported from ticks (Appendix A). The main *Haemaphysalis* species reported was *H. punctata*, which was reported for 11 TBP species of the 21 reported in the genus. It was one of the main tick species in Slovenia (Figure 3). This species is mainly responsible for the distribution of parasites from the genera *Babesia* and *Theileria*, but can also be positive for a large panel of bacteria in wild animals [168] (Appendix A).



**
*Hyalomma*
**



The *Hyalomma* genus is well known for its transmission of CCHFV. They are large-sized ticks with a tropism for large mammals in the case of adult forms, or for birds and small mammals in the immature stages. This genus is mainly distributed in the southern part of the Mediterranean Rim; their sparse distribution in the northern part is, however, increasing [168,169]. A total of six species (*Hy. aegyptum*, *Hy. anatolicum excavatum*, *Hy. detritum*, *Hy. lusitanicum*, *Hy. marginatum*, and *Hy. rufipes*) were collected from 12 wild animals. This genus was found in the greatest number of countries (Algeria, Cyprus, Greece, France, Israel, Italy, Morocco, Palestine, Spain, and Turkey). It was found positive for 26.2% of the TBPs recorded in ticks. The main *Hyalomma* species reported for the largest number of TBPs was *Hy. marginatum* (Appendix A). A total of 13 TBPs were found in this particular species of the 16 species found in the genus, and it was defined as a main tick species in Greece and Morocco (Figure 3). *Hyalomma marginatum* was reported to be found on wildlife in many different countries. It is of importance in the area of healthcare because it can transmit TBPs such as *Rickettsia* spp., CCHFV, *Babesia* spp., and *Theileria* spp. [172]. In combination with the immature stage’s tropism for birds, these characteristics could warrant the importance given to this species in health terms [173]. This confirms the importance of *Hy. marginatum* in TBP monitoring. However, like the genus *Ixodes*, particular attention was given to other species such as *Hy. aegyptium* in Algeria and Palestine, and *Hy. rufipes* in Greece (Figure 3).



**
*Dermacentor*
**



*Dermacentor* ticks mainly target mammals and are found more or less worldwide. The genus is mainly represented by two species commonly found in the Mediterranean Rim: *D. marginatus* and *D. reticulatus* [168]. They transmit a wide range of TBPs, including *Rickettsia* spp. and piroplasmids, making them important from both a veterinary and human health viewpoint [168]. *Dermacentor marginatus* on wild animals was found positive for 12 TBPs in five Mediterranean Rim countries (Algeria, France, Italy, Slovenia, and Spain). It was defined as a main tick species in Slovenia (Appendix A, Figure 3). All the stages were reported to have been found on ten wildlife species, but it is mainly found on wild ungulates such as wild boars. It is responsible for the circulation of TBPs from the genera *Babesia* and *Rickettsia* [174]. Its main tropism for wild ungulates during the adult stage and micromammals and birds for the immature stages could explain the numerous data reported for this species.



**
*Amblyomma*
**



Ticks in the genus *Amblyomma* transmit pathogens responsible for diseases such as Rickettsiosis, and are associated with almost all terrestrial animals. They are mainly reported in tropical and sub-tropical areas [175,176,177]. The genus was poorly represented in the Mediterranean Rim. It was found at the generic level in two countries (Greece and Israel). The *Am. marmoreum* species was found in Italy on (mainly migratory) birds, and two bacteria genera were detected: *Erhlichia* spp. (*R. aeschlimannii*) and *Rickettsia* spp. (Appendix A).

### 5.2. Argasidae



**
*Ornithodoros*
**



*Ornithodoros* is involved in the transmission of diseases such as CCHF and relapsing fever in the Middle East and Africa [178,179]. In the Mediterranean Rim, it was collected from two wildlife species in three countries (Morocco, Spain, and Tunisia). Three *Ornithodoros* species were reported (*O. erritacus*, *O. maritimus*, and *O. normandi*) with a prevalence of 8.2% of the TBPs detected in ticks. The tick species with the highest number of TBPs (three) was *O. erritacus* (Appendix A). This could be explained by its main tropism for rodents and micro mammals [168]. However, it was not defined as a main tick species, unlike *O. normandi*, which was defined as a main tick species in Tunisia (Figure 3).



**
*Argas*
**



The genus *Argas* is reported in the Mediterranean Rim but more especially involved in the circulation of diseases related to birds and bats [169]. It was the least reported tick genus, with only two species mentioned: *A. transgariepinus* (one TBP reported) and *A. vespertilionis* (two TBPs reported). It was found only in Italy on bats and involved in the detection of two bacterial genera: *Bartonella* spp. and *Rickettsia* spp. Along with the genus *Amblyomma* (hard ticks), *Argas* is the least collected genus in the Mediterranean Rim (Appendix A).

## 6. Wild Animal Hosts of TBP-Positive Ticks and Harboring TBPs in Mediterranean Rim Countries

This section focuses on data related to wild animals infected by TBPs and the wild animal hosts of TBP-positive engorged ticks. Nevertheless, the data on the latter do not confirm the ticks’ vector character for the different TBPs.

A total of 17 wild animal groups that hosted both ticks and pathogens were reported in our data. From these wild animals, 61 TBPs were reported as being detected from ticks and 50 TBPs were reported as being detected directly from wild animals (Appendix A). The complete list of TBPs reported in our study from TBP-positive engorged ticks on wild animals and from TBP-positive wild animals is shown in Table 2. In this section, the wild animal harboring TBPs reported in the most publications in a considered country was defined as a main TBP host in that country, and the wild animal reported as a host of TBP-positive engorged ticks in the most publications in a considered country was similarly defined as a main host of TBP-positive engorged ticks in that country. The main groups reported positive for TBPs were rodents, in which 46% of the TBPs in wild animals were reported; they were therefore defined as main TBP hosts in Croatia, France, Italy, Israel, and Morocco. This group was followed by roe deer (20%) and red foxes (16%), both defined as main TBP hosts in Turkey. Next came wild boars (14%), defined as a main TBP host in Algeria and Turkey, then wild carnivores (14%). Red deer (10%) were defined as a main host in Spain, Turkey, and Slovenia, while birds (10%) were defined as a main host in Cyprus and Turkey. Those found to have the lowest percentage of TBPs were mouflons (6%, defined as a main host in Cyprus), chamois (6%), hares (4%, defined as a main host in Turkey), lizards (4%, defined as a main host in Palestine), tortoises (4%, defined as a main host in Palestine and Turkey), swamp deer (4%), fallow deer (2%), and porcupines (2%) (Appendix A).

The main wild animal groups reported to have the highest diversity of TBPs in TBP-positive ticks were small wild animals such as rodents (36.1%) and birds (34.4%), followed by game mammals such as wild boars (27.9% of TBPs), red deer (26.2%), and roe deer (22.9%) (Table 3). The amount of data found on rodents and birds can be explained by their role as hosts of immature ticks and the potential role of migratory birds in tick dissemination [12,180]. The game mammal hosts are well-known game animals that can also interact with livestock. These interactions—combined with the accessibility of samples due to their game status—may explain the scientific interest in these wild animals [181,182]. They were followed by red foxes (21.3%) and wild carnivores such as badgers, genets, golden jackals, lynxes, martens, mongooses, otters, and wolves (13.1%). Concerning wolves, data were sporadically obtained from animal captures for veterinary purposes or from samples taken directly from the corpse after natural or accidental death. The remaining wild ungulates reported were composed of species such as mouflons (18%), fallow deer (16.4%), and chamois (3.3%), which are also game mammals and probably the focus of research for the same reason as red deer and wild boars, though they are of less scientific interest. The final group of wild animal hosts comprises the hosts of immature ticks, which include lizards (14.8%), hedgehogs (9.9%), tortoises (9.9%), hares (8.2%), and bats (6.6%) (Table 2). They are also hosts of specific tick species such as *Hy. aegyptium* on tortoises. Some wild animals were hosts to ticks with a high percentage of TBPs, and birds were defined as the main host for TBP-positive engorged ticks in many (five) countries. This could be explained by the scientific interest in the role of birds in the tick’s life cycle and their dissemination of TBPs [12,173].

## 7. Biogeography, Diversity, and Distribution in the Mediterranean Rim

The distribution of publications throughout Mediterranean Rim countries is shown in Figure 4A. This gives a total of 148 publications in 22 countries, with a mean number of seven articles per country. We can note a variability in these data depending on their geographical situation. The number of publications can vary from 0 to 54 according to the country. Most of the articles included in our review concern Italy (54), Spain (25), and France (23). The remaining publications originated from the other countries of the Mediterranean Rim, with the exception of seven countries where no publications were included in our dataset (Albania, Egypt, Lebanon, Libya, Malta, Montenegro, and Syria). However, if it seems to show that the most important research efforts are in Italy, France, and Spain, this is not necessarily true. Along with the country, the financial support, the political situation, the accessibility to the wild animal host due to their protection status, and local research habits could also impact the performance of research efforts in certain countries.

### 7.1. Overall Analysis of the Four Areas of the Mediterranean Rim

For the purposes of this review, we divided the Mediterranean Rim countries into four areas: Western Europe, which includes France, Italy, Malta, Monaco, and Spain (68.8% of the publications); the Balkans, composed of Albania, Bosnia-Herzegovina, Croatia, Greece, Montenegro, and Slovenia (8.8% of the publications); the Middle East, composed of Cyprus, Israel, Lebanon, Palestine, Syria, and Turkey (15.5% of the publications); and North Africa, which includes Algeria, Egypt, Libya, Morocco, and Tunisia (6.8% of the publications) (Figure 4A). 



**Western Europe**



In this review, Western Europe was divided into five countries: France, Italy, Malta, Monaco, and Spain. Publications about pathogens in wild animals were only found in France, Italy, and Spain. These countries represented the highest amount of research in the Mediterranean Rim, having produced 102 papers, corresponding to 68.9% of all the publications included. Among the 17 animal hosts reported in this review, 16 were found in Western Europe. Of these, five were often included in the publications of this area: red deer (32%), wild boars (27%), rodents (26%), birds (25%), and roe deer (17%) (Figure 4B). A total of 31 tick species were found on these 16 hosts in Western Europe, which corresponds to 97% of the total number of tick species found in our review of Mediterranean Rim countries. The seven tick genera reported in this area include *Ixodes*, a ubiquitous tick genus, followed by the genera *Hyalomma* and *Dermacentor*, both targeting ungulates [168,169]. The high prevalence of *Ixodes* spp. in Western Europe compared to the other three areas is due to its range, which is mainly in Europe [104]. The high prevalence of *Hyalomma* could be due to the recent expansion of this genus from other areas [16,108]. The immature stages of these three genera can be found on animal hosts such as rodents and birds [110,180]. The other tick genera reported were *Argas* spp., *Amblyomma* spp., *Haemaphysalis* spp., *Orniyhodoros* spp., and *Rhipicephalus* spp. Among the TBPs reported from these wild animals and their ticks, the three most frequently found pathogens were *A. phagocytophilum*—the causative agent of human granulocytic anaplasmosis (HGA)—which was found in more than 30% of the publications in Western Europe, and bacteria from the genus *Rickettsia*, including *R. slovaca*, a bacterium from the spotted fever group (SFG) found in 14% of the publications, *R. helvetica*, reported in 13% of the publications, and *C. burnetti*, also found in 13% of the publications (Figure 4C). In Western Europe, the data reported were mainly from research focusing on game animals such as wild ungulates (red deer and wild boars) and their ticks. They also focused on animal hosts of immature ticks (rodents).



**Middle East**



The Middle East is the second most important area in terms of publications, with 23 publications on wild animals (15.5% of the total amount of publications included). These publications reported 12 animal hosts, which corresponds to 70.5% of the animal hosts found in the Mediterranean Rim. Of all these animal hosts, five were found more frequently. These were tortoises, birds, rodents, red deer, and wild boars, reported in 29.2%, 16.7%, 16%, 16%, and 12.5% of publications from the Middle East, respectively (Figure 4B). Twenty-nine tick species from six genera were reported to have been collected from these animal hosts, representing 90.6% of the ticks found in the Mediterranean Rim. The main genera reported were *Hyalomma* spp., followed by *Rhipicephalus* spp., *Haemaphysalis* spp., *Ixodes* spp., *Dermacentor* spp., and *Argas* spp. Thirty-nine pathogens were detected in wild animals and their ticks (51.3% of the total number of pathogens). Of these, the most reported in terms of the percentage of publications from this area were *Rickettsia* spp., a genus responsible for diseases such as spotted fever and TIBOLA (20.8%) [162]; CCHFV, mainly transmitted by *Hyalomma* ticks (16.7%); R. massiliae, an SFG bacterium (12.5%) (Figure 4C); and *R. aeschlimannii*, and *C. burnetiid*, both reported in 8.3% of the area’s publications. Unlike Western Europe, the scientific focus appeared to be not only on game mammals—although red deer were often the subject of research, other wild animal hosts involved in the tick’s life cycle and dispersion were also investigated, such as birds and tortoises [110].



**The Balkans**



A total of 13 publications were included in the Balkan area, representing 8.8% of all the publications (Figure 4A). Eight animal hosts were reported to be infected in these papers. The main five wild animals reported were red deer, rodents, wild boars, roe deer, and birds, mentioned in 26.7%, 26.7%, 13.3%, 13.3%, and 6.7% of the area’s publications, respectively (Figure 4B). A total of 16 tick species from six genera were reported from these hosts (51.6% of the total tick species found). The main genera reported was *Hyalomma* spp., followed by *Rhipicephalus* spp., *Haemaphysalis* spp., *Ixodes* spp., *Argas* spp., and *Ornithodoros* spp. A total of 16 pathogens were found in wild animals and their ticks in the Balkans, which represents 21.1% of the total pathogens found in this review. The five main pathogens found were *A. phagocytophilum*, found in 20% of the publications; the genus *Babesia*, found in 20% of the publications; TBEV, found in 7.5% of the publications; and the genera *Leptospira* and *Rickettsia*, which were both reported in 13.3% of the area’s publications (Figure 4C). Like Western Europe, the scientific focus was on game animals such as red deer and wild boars, in addition to micro mammals.



**North Africa**



The Mediterranean Rim area with the fewest number of publications found was North Africa, with ten publications. This area represents 6.8% of all the publications from the Mediterranean Rim (Figure 4A). A total of six wild animal hosts were reported in these studies, which represents about 35.3% of all the hosts found. The main wildlife hosts in this area were tortoises and hedgehogs, which were both reported in 30% of the papers from North Africa. Birds, rodents, and wild boars were each reported in 20% of the papers in North Africa (Figure 4B). Eighteen tick species from eight tick genera found on these six hosts were reported as TBP-positive (about 58% of the tick species reported in this review). Of these, the main species reported was also *Hyalomma* spp., followed by *Rhipicephalus* spp., *Ixodes* spp., *Haemaphysalis* spp., *Argas* spp., *Amblyomma* spp., *Dermacentor* spp., and *Ornithodoros* spp. Eleven pathogens were reported from these ticks and animal hosts. This represents 14.5% of the total pathogens found in the review. The most frequently found in North Africa were *R. aeschlimannii*, an SFG rickettsia found in 30% of the publications in North Africa; *Borrelia* spp.; CCHFV; and *R. massiliae* and *R. slovaca*, each of which was reported in 20% of the publications (Figure 4C). North Africa has the fewest number of publications in the Mediterranean Rim, with a main focus on hosts of immature ticks such as birds and hedgehogs, though there is a secondary focus on hosts of the Hyalomma tick, such as tortoises.

### 7.2. Insular Tick-Borne Pathogens in Wild Animals and Their Ticks

Due to their geographic situation, the Mediterranean islands are of importance in terms of animal migrations, human activities, and pathogen circulation. A total of 19 papers were published from these islands, representing 12.8% of the publications selected during this review. These 19 publications were included in the complete dataset. From 19 papers, most were from some of the largest islands in the Mediterranean Rim: Sardinia, Corsica, Cyprus, and smaller Greek islands. We decided to split the islands according to their geographic location between the western and eastern parts of the Mediterranean Sea. The western islands include Asinara, Capris, Columbretes, Corsica, Isla Grossa, Majorca, Medes, Pianosa, Ponza, Sardinia, Sicily, Ustica, Ventotene, and Zanone (16 of the publications), while the eastern islands include Antikythira and Cyprus (four of the publications) (Table 3). One publication was reported from both western and eastern islands.

A total of 11 wild animal hosts were reported to carry TBP-positive ticks or be infected directly by TBPs: birds, hares, hedgehogs, foxes, mouflons, porcupines, red deer, rodents, tortoises, wild boars, and wild carnivores (Table 3). However, most of the data were reported in publications about TBPs in TBP-positive ticks collected from these wild animals (89.5% of the island publications). In the western islands, 10 of the 11 wild animals were reported: birds, hedgehogs, mouflons, porcupines, red deer, red foxes, rodents, tortoises, wild boars, and wild carnivores. Of this number, most of the data (62.5%) in the western island publications were about birds. They were closely followed by wild boars, which were reported in 50% of the publications, then mouflons in 31.3%. The last two main wild animals reported were red deer and wild carnivores, both reported in 18.8% of the western island publications (Figure 4D). Meanwhile, fewer wild animals (four) were reported in the eastern island publications: birds and mouflons were both reported in 50% of them, and hares and red foxes in 25% of them (Figure 4D).

Concerning TBP-positive ticks, of the eight genera reported in the complete dataset, seven were reported in the island dataset: *Amblyomma* spp., *Dermacentor* spp., *Haemaphysalis* spp., *Hyalomma* spp., *Ixodes* spp., *Ornithodoros* spp., and *Rhipicephalus* spp. (Table 3). The seven genera were reported in the western island dataset. The main TBP-positive tick genus reported was *Hyalomma* spp. in 64.3% of publications, followed by the genera *Rhipicephalus* and *Ixodes*, reported in 57.1% and 50% of the publications on ticks, respectively. This is probably due to the range of *Hyalomma* spp. and their spread northward within the Mediterranean Rim [16,173]. This genus may thus contribute to the extension of TBP circulation, particularly on the islands, which can play a sentinel role [108]. Indeed, birds are the main wild animal hosts of *Hyalomma’s* immature stage on the islands. The last two main genera were *Dermacentor* and *Haemaphysalis*, both reported in 35.7% of the TBP-positive tick publications from the western islands (Figure 4D). As for the wild animal hosts, fewer positive tick genera were reported in the eastern island dataset (five): *Haemaphysalis*, *Hyalomma*, and *Rhipicephalus* were found in 100% of the publications on TBP-positive ticks, followed by *Ixodes* spp. and *Amblyomma* spp. reported in 66.7% and 33.3% of the eastern island publications concerning TBP-positive ticks, respectively (Figure 4D). The pattern of the main TBP-positive tick genera reported in the island dataset followed the same pattern as the complete dataset, with the exception of Western Europe.

Of all the TBPs reported, 36.8% of them were identified in the TBP-positive wild animals and their TBP-positive ticks on the islands (Table 3). Even though a small percentage of the complete dataset was reported in the island publications, a relatively high percentage of the TBPs was reported. On the western islands, a total of 11 genera were reported: *Anaplasma* spp., *Babesia* spp., *Bartonella* spp., *Borrelia* spp., CCHFV, *Chlamydia* spp., *Coxiella* spp., *Ehrlichia* spp., *Maebian* virus, *Rickettsia* spp., and *Theileria* spp. Of these, the main genus reported was *Rickettsia* spp. in 68.8% of the publications from the western islands, followed by CCHFV and the genera *Anaplasma*, Ehrlichia, and *Coxiella* reported in 12.5%, 18.8%, 12.5%, and 6.3% of the western island publications, respectively (Figure 4D). Fewer TBPs (three) were reported in the eastern island publications: *Rickettsia* spp., *Anaplasma* spp., and *Coxiella* spp. were reported in 100%, 50%, and 50% of these publications, respectively (Figure 4D).

## 8. Conclusions

A total of 148 publications reported TBPs in wild animals and their ticks. Of these, 68.9% were reported in Western Europe, 15.5% in the Middle East, 8.8% in the Balkans, and 6.8% in North Africa. This shows that Western Europe’s scientific research mainly focuses on TBPs in wild animals within Western Europe compared to the other areas investigated. Additionally, most of the research was performed in four countries: France, Italy, Spain, and Turkey; most of the Mediterranean Rim remains unexplored. These publications reported 85 pathogens, with nine genera for bacteria, eight for parasites, and four for viruses, revealing the marked diversity of TBPs and the importance of the research on TBPs in wild animal populations. The main TBP-positive genus reported was *Ixodes* spp., detected positive for more than 60% of the TBPs. However, this was the main TBP-positive genus reported in Western Europe alone. The importance of this genus in the complete dataset could be due to the greater number of publications from Western Europe. In the other areas and on the Mediterranean islands, the main TBP-positive tick genus was *Hyalomma* spp. The numerous data found on *Hyalomma* ticks could be explained by both the amount of data reported on their wild animal hosts, such as wild ungulates and birds, and the recent expansion of this genus repartition area, which led to the expansion of the reparation area of the TBPs this genus carries. The 17 wild animal hosts reported in the complete dataset included a large diversity of wild animal groups, namely wild ungulates such as red deer and wild boar, which host adult ticks, and micro-vertebrates such as birds and rodents, which host immature ticks. The amount of data about birds on the islands (63.2% of island publications) could be due to the potential importance of migration paths in the dissemination of ticks and TBPs there. This shows the importance of wildlife in the circulation of tick-borne pathogens. The main deficiency of this review of tick-borne pathogens in wild animals and their tick species is that the Mediterranean islands only provided 12.8% of the total publications, even though 36.8% of the pathogens and nearly 64.7% of the wild animal hosts and tick species were found there.

## Figures and Tables

**Figure 1 microorganisms-10-01858-f001:**
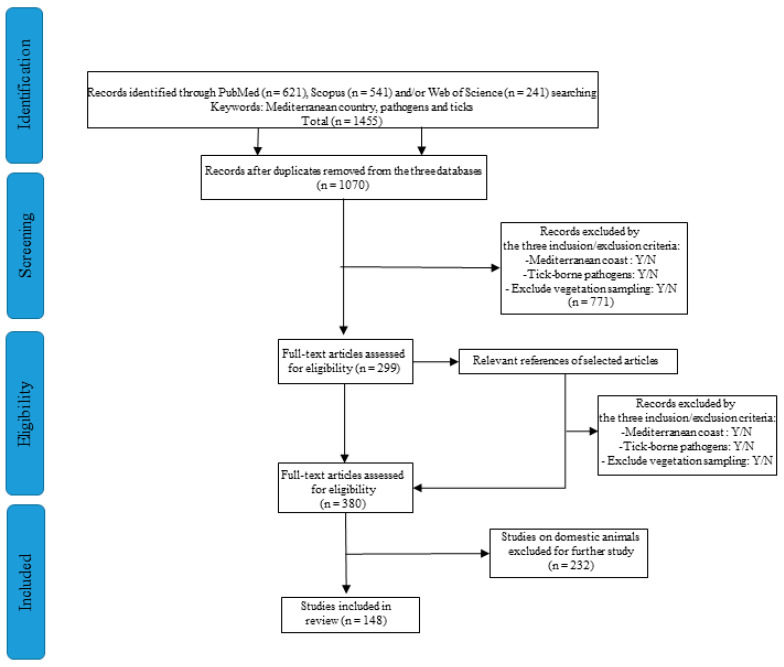
Methodology diagram for the bibliographic research in keeping with the PRISMA 2009 flow chart according to Moher et al., 2015.

**Figure 2 microorganisms-10-01858-f002:**
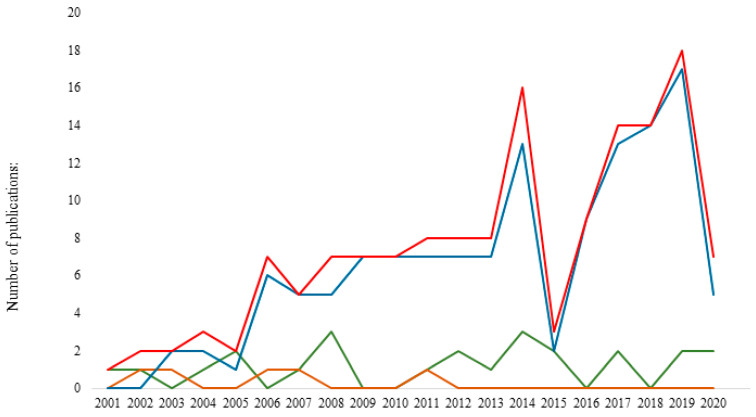
Number of publications through the years (2000–February 2021). The total number is indicated in red, and the other colored lines indicate the number of publications according to each detection method used (serology, microscopy, or molecular biology).

**Figure 3 microorganisms-10-01858-f003:**
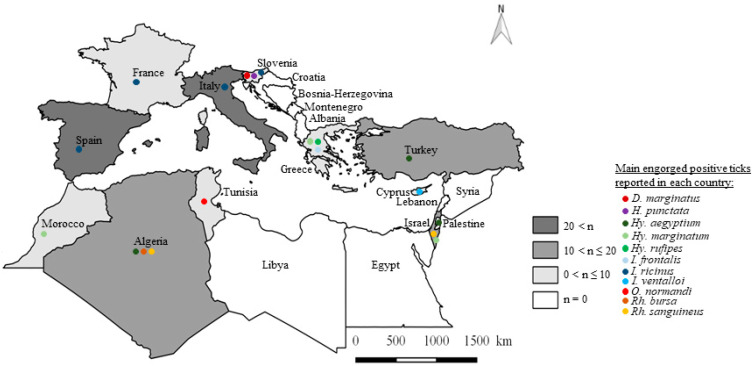
Distribution of main TBP-positive engorged tick species in Mediterranean Rim countries (n = number of TBP-positive engorged tick species by country).

**Figure 4 microorganisms-10-01858-f004:**
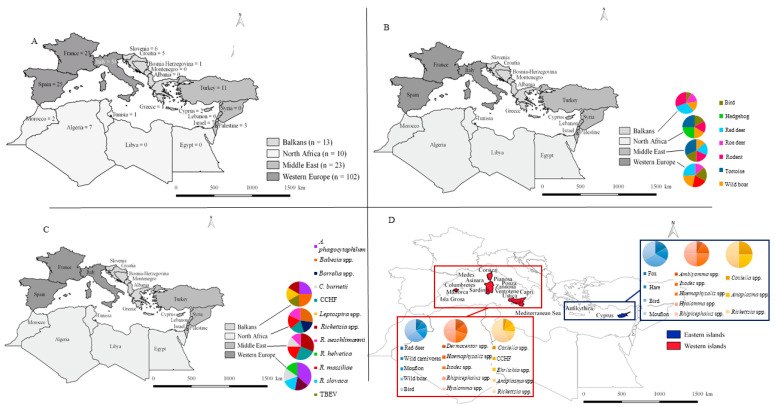
(**A**) Map of publications per country and Mediterranean area (n = number of publications), (**B**) of the main wild animal hosts, (**C**) of the main TBPs in the four Mediterranean Rim areas, (**D**) and main wild animal hosts, genera of TBP-positive ticks, and TBPs on the western and eastern Mediterranean islands.

**Table 1 microorganisms-10-01858-t001:** Tick-borne pathogens reported in wild animals or engorged ticks found on wild animals in Mediterranean Rim countries.

Pathogen	TBP-Positive Engorged Ticks Collected from Hosts	Hosts of TBP-Positive Ticks	TBP-Positive Hosts	Country	Reference
**Parasites: Nematoda**					
** *Cercopithifilaria* **					
*Cercopithifilaria rugosicauda*	No data found	No data found	Roe deer	Italy	[19]
**Parasites: Apicomplexa**					
** *Babesia* **					
*Babesia spp.*	*Dermacentor marginatus*, *Hyalomma* spp., *Hyalomma marginatum*, *Ixodes ricinus*, and *Rhipicephalus bursa*	Deer, roe deer, and wild boars	Deer, badgers, hares, roe deer, rodents, and wolves	Croatia, France, Italy, and Spain	[22,23,24,25,26,27,28,29]
*Babesia annae*	No data found	No data found	Red foxes	Italy	[21]
*Babesia bigemina*	No data found	No data found	Red deer, roe deer, and wild boars	Italy and Spain	[26,30]
*Babesia canis*	No data found	No data found	Wolves	Croatia	[31]
*Babesia capreoli*	*Ixodes ricinus*	Birds	Chamois, red deer, and roe deer	Italy and Spain	[26,30,32]
*Babesia crassa*	*Haemaphysalis parva*	Wild boars	No data found	Turkey	[25]
*Babesia divergens*	No data found	No data found	Red deer	Italy	[33]
*Babesia microti*	*Ixodes hexagonus* and *Ixodes ricinus*	Red foxes, rodents, and roe deer	Red foxes and rodents	Croatia, Italy, and Turkey	[20,21,32,34,35]
*Babesia occultans*	*Hyalomma marginatum*	Wild boars	No data found	Turkey	[25]
*Babesia ovis*	*Ixodes ricinus*	Roe deer	No data found	Italy	[24]
*Babesia rodaini*	*Ixodes ricinus* and *Rhipicephalus sanguineus* s.l.	Roe deer and wild boars	No data found	Italy	[24]
*Babesia rossi*	*Haemaphysalis parva*	Wild boars	No data found	Turkey	[25]
*Babesia venatorum*	*Ixodes ricinus*	Birds and rodents	Roe deer	Italy and Spain	[26,30,32]
*Babesia vulpes*	No data found	No data found	Red foxes	Italy and Turkey	[21,25,28]
** *Cytauxzoon* **					
*Cytauxzoon* spp.	No data found	No data found	Lynxes	Spain	[36]
** *Hemolivia* **					
*Hemolivia mauritanica*	No data found	No data found	Tortoises	Palestine	[37]
** *Hepatozoon* **					
*Hepatozoon* spp.	No data found	No data found	Lizards	Palestine	[38]
*Hepatozoon canis*	No data found	No data found	Red foxes	Croatia, Israel, and Italy	[39,40,41]
*Hepatozoon kisrae*	No data found	No data found	Lizards	Palestine	[38]
** *Theileria* **					
*Theileria* spp.	*Dermacentor marginatus*, *Hyalomma marginatum*, *Ixodes ricinus*, and *Rhipicephalus bursa*	Fallow deer, red deer, and wild boar	Chamois, hares, fallow deer, red deer, roe deer, and wild boars	Italy and Spain	[23,26,27,30,42,43]
*Theileria annae*	*Ixodes hexagonus* and *Ixodes ricinus*	Red foxes and rodents	Badgers and red foxes	Croatia, Italy, and Spain	[21,30,39,40,44,45]
*Theileria buffeli*	*Hyalomma marginatum*, *Rhipicpehalus bursa*, and *Rhipicephalus sanguineus* s.l.	Birds, red foxes, and mouflons	No data found	Italy	[46]
*Theileria capreoli*			Wolves	Croatia	[31]
*Theileria orientalis*	*Hyalomma marginatum*, *Rhipicephalus bursa*, and *Rhipicephalus sanguineus* s.l.	Birds, red foxes, and mouflons	No data found	Italy	[46]
*Theileria ovis*	*Rhipicephalus ovis*	Mouflons	Chamois	Italy and Spain	[43]
*Theileria sergenti*	*Hyalomma marginatum*, *Rhipicephalus bursa*, and *Rhipicephalus sanguineus* s.l.	Birds, red foxes, and mouflons	No data found	Italy	[46]
**Bacteria**					
** *Anaplasma* **					
*Anaplasma* spp.	*Haemaphysalis punctata*, *Ixodes ricinus*, *Rhipicephalus bursa*, and *Rhipicephalus sanguineus* s.l.	Chamois and hedgehogs	Birds	Algeria, Cyprus, and Italy	[47,48,49]
*Anaplasma capra*	No data found	No data found	Red deer and swamp deer	France	[50]
*Anaplasma marginale*	*Dermacentor marginatus*, *Hyalomma marginatum*, *Ixodes ricinus*, and *Rhipicephalus bursa*	Red deer and wild boar		Italy	[23]
*Anaplasma phagocytophilum*	*Boophilus kohlsi*, *Dermacentor marginatus*, *Haemaphysalis punctata*, *Hyalomma marginatum*, *Ixodes ricinus*, *Rhipicephalus bursa*, and *Rhipicephalus sanguineus* s.l.	Bird, fallow deer, red deer, roe deer, rodent, and wild boar	Fallow deer, porcupines, red foxes, roe deer, rodents, swamp deer, and wild boars	France, Greece, Israel, Italy, Slovenia, and Spain	[21,23,32,33,40,50,51,52,53,54,55,56,57,58,59,60,61,62,63,64,65,66,67,68,69]
*Anaplasma ovis*	*Haemaphysalis punctata*, *Haemaphysalis sulcata*, *Ixodes festai*, *Rhipicephalus bursa*, and *Rhipicephalus sanguineus* s.l.	Hedgehog, marten, mouflon, and red fox	Mouflons	Cyprus and Italy	[70,71]
** *Bartonella* **					
*Bartonella* spp.	*Argas vespertilionis*, *Ixodes vespertilionis*, *Ixodes simplex*,, and *Rhipicephalus sanguineus* s.l.	Bats, red fox	Birds and rodents	Israel and Italy	[70,72,73,74]
*Bartonella acomodis*	*Ixodes ricinus*	Red deer	No data found	Italy	[59]
*Bartonella bacilliformis*	*Haemaphysalis punctata* and *Ixodes ricinus*	Fallow deer and red deer	No data found	Italy	[59]
*Bartonella bovis*	*Dermacentor marginatus*, *Haemaphysalis punctata*, and *Ixodes ricinus*	Fallow deer and red deer	No data found	Italy	[59]
*Bartonella chomelii*	*Dermacentor marginatus* and *Ixodes ricinus*	Red deer and roe deer	No data found	Italy	[59]
*Bartonella elizabethae*	No data found	No data found	Rodent	Israel	[74]
*Bartonella tamiae*	*Ixodes vespertilionis*	Bats	No data found	Algeria	[75]
*Bartonella tribocorum*	*Ixodes ricinus*	Red deer	No data found	Italy	[59]
*Bartonella vinsonnii berkoffii*	*Ixodes ricinus*	Red deer	No data found	Italy	[59]
** *Borrelia* **					
*Borrelia* spp.	*Hyalomma aegyptium*	Tortoises	Rodents	Algeria, Morocco, Spain, and Turkey	[48,76,77,78]
*Borrelia afzelii*	*Ixodes ricinus*	Birds, lizards, and rodents	Rodents	France, Italy, Spain, and Turkey	[11,76,79,80,81,82,83,84]
*Borrelia bavariensis*	No data found	No data found	Rodents	France	[81]
*Borrelia burgdoferi* s.l.	*Dermacentor marginatus*, *Ixodes acuminatus*, *Ixodes ricinus*, and *Rhipicephalus sanguineus* s.l.	Fallow deer, lizards, red deer, rodents, and roe deer	Birds, rodents, roe deer, and wild boars	France, Italy, and Spain	[33,58,59,64,72,76,81,83,84,85,86,87]
*Borrelia burgdoferi* s.s.	*Ixodes acuminatus* and *Ixodes ricinus*	Rodents and roe deer	Rodents	France, Italy, and Spain	[79,81,82,83,88,89]
*Borrelia crocidurae*	*Ornithodoros erritacus*	Rodents	No data found	Morocco	[77]
*Borrelia garinii*	*Haemaphysalis punctata*, *Ixodes* spp., *Ixodes frontalis*, and *Ixodes ricinus*	Birds, lizards, and rodents	Rodents	France, Italy, and Spain	[11,63,79,82,83,84]
*Borrelia hispanica*	*Ornithodoros erritacus*	Rodents	No data found	Morocco	[77]
*Borrelia lusitaniae*	*Ixodes ricinus*	Bird, lizards, rodents, and roe deer	Rodents	Italy	[9,11,83,84,88,90,91]
*Borrelia merionesi*	*Ornithodoros erritacus*	Rodents	Rodents	Morocco	[77]
*Borrelia spielmanii*	*Ixodes acuminatus*	Birds	No data found	Italy	[92]
*Borrelia valaisiana*	*Ixodes* spp., *Ixodes ricinus*, and *Ixodes ventalloi*	Birds, lizards, and rodents	Rodents	Italy and Spain	[11,63,79,83,84,90,92]
*Borrelia turdi*	*Ixodes frontalis*	Birds	No data found	Spain	[63]
** *Chlamydia* **					
*Chlamydia* spp.	No data found	No data found	Birds	Italy	[72]
*Chlamydia abortus*	*Haemaphysalis sulcata*, *Rhipicephalus bursa*, and *Rhipicephalus sanguineus* s.l.	Hedgehog, marten, mouflon, red deer, and red fox	No data found	Italy	[93]
*Chlamydia psitacci*	No data found	No data found	Bird	Italy	[72]
** *Coxiella* **					
*Coxiella burnetii*	*Haemaphysalis punctata*, *Haemaphysalis sulcata*, *Hyalomma anatolicum excavatum*, *Ixodes* spp., *Ixodes acuminatus*, *Ixodes ricinus*, *Ixodes ventalloi*, *Ixodes vespertilionis Rhipicaphalus bursa*, and *Rhipicephalus sanguineus* s.l.	Bat, bird, hare, marten, mouflon, roe deer, rodent, and wild boar	Bird, mouflon, red deer, red fox, and rodent	Algeria, Cyprus Italy, Slovenia, and Spain	[33,40,49,58,70,71,72,75,76,86,94,95,96,97]
** *Ehrlichia* **					
*Ehrlichia* spp.	*Am. marmoreum*, *Dermacentor marginatus*, *Hyalomma* spp., *Hyalomma marginatus*, *Hyalomma rufipes*, and *Rhipicephalus bursa*	Bird, red deer, and wild boar		Spain	[23,98]
*Ehrlichia canis*	*Dermacentor marginatus*, *Haemaphysalis punctata*, and *Rhipicephalus sanguineus* s.l.	Mouflon, red fox, and wild boar	Otter, red fox, and wolf	Israel and Italy	[40,41,70,99,100,101]
** *Leptospira* **					
*Leptospira* spp.	No data found	No data found	Rodent	Croatia	[29]
*Leptospira borpetersenii*	No data found	No data found	Rodent	Croatia, France	[102,103]
*Leptospira interrogans*	No data found	No data found	Rodent	Croatia, France	[29,102,103]
*Leptospira kishneri*	No data found	No data found	Rodent	Croatia	[103]
*Leptospira santarorai*	No data found	No data found	Rodent	Croatia	[103]
** *Noehrlichia* **					
*Neoehrlichia mikurensis*	*Ixodes ricinus*	Rodent	Rodent	France and Italy	[32,104]
** *Rickettsia* **					
*Rickettsia* spp.	*Argas transgariepinus*, *Argas vespertilionis*, *Dermacentor marginatus*, *Haemaphysalis spp*., *Haemaphysalis puntata*, *Haemaphysalis sulcata*, *Hyalomma aegyptium*, *Hyalomma marginatum*, *Ixodes frontalis*, *Ixodes ricinus*, *Ixodes ventalloi*, *Rhipicephalus bursa*, *Rhipicephalus pusillus*, *Rhipicephalus sanguineus s.l.*, and unknown	Bat, bird, hare, hedgehog, lizard, mouflon, red deer, roe deer, turtle, tortoise, and wild boar	Bird, mouflon, and otter	Algeria, Cyprus, Italy, Spain, and Turkey	[23,47,48,49,58,63,71,72,73,84,99,101,105,106,107,108]
*Rickettsia aeschlimannii*	*Am. marmoreum*, *Haemaphysalis* spp., *Hyalomma* spp., *Hyalomma aegyptium*, *Hyalomma marginatum*, *Hyalomma rufipes*, *Ixodes frontalis*, and unknown	Bird, roe deer, tortoise, and wild boar	No data found	France, Greece, Israel, and Italy	[98,108,109,110,111,112,113,114,115]
*Rickettsia africae*	*Hyalomma* spp., *Hyalomma aegytium*, *Hyalomma detritum*, *Hyalomma marginatum*, *Hyaloma rufipes*, *Ixodes ricinus*, and unknown	Bird, tortoise, and wild boar	No data found	Algeria, Greece, Israel, Italy, and Turkey	[48,98,108,109,110,116,117]
*Rickettsia helvetica*	*Ixodes acuminatus*, *Ixodes festai*, *Ixodes ricinus*, and *Ixodes ventalloi*	Bird, chamois, fallow deer, hedgehog, lizard, lynx, red deer, red fox, roe deer, and rodent	No data found	France, Italy, and Spain	[11,32,63,70,84,92,106,118,119]
*Rickettsia hoogstraalii*	*Haemaphysalis punctata*, *Haemaphysalis sulcata*, and *Ixodes ricinus*	Lizard and mouflon	No data found	Cyprus and Italy	[70,84,120,121]
*Rickettsia massiliae*	*Haemaphysalis erinacei*, *Rhipicephalus pusillus*, and *Rhipicephalus sanguineus* s.l.	Hare, hedgehog, golden jackal, lynx, mongoose, red fox, roe deer, and wild boar	No data found	Algeria, Cyprus, Israel, Italy, and Spain	[70,75,113,117,118,119,120,121,122]
*Rickettsia monacensis*	*Haemaphysalis punctata*, *Ixodes ricinus*, and *Ixodes ventalloi*	Badger, bird, fallow deer, genet, hare, lizard, lynx, mongoose, red deer, red fox, roe deer, and rodent	No data found	Spain and Italy	[11,32,59,63,83,84,91,106,118,119]
*Rickettsia raoultii*	*Dermacentor marginatus* and *Hyalomma* spp.	Bird, fallow deer, rodent, and wild boar	No data found	Italy and Spain	[83,106,110,118,123]
*Rickettsia sibirica*	*Ixodes ricinus*	Bird	No data found	Spain	[63]
*Rickettsia sibirica mongolitimonae*	*Rhipicephalus (Bo.) annulatus*	Fallow deer	No data found	Israel	[117]
*Rickettsia slovaca*	*Dermacentor marginatus*, *Haemaphysalis punctata*, *Ixodes ricinus*, and *Rhipicephalus bursa*	Fallow deer, red deer, rodent, and wild boar	Rodent and Wild boar	Algeria, France, Italy, and Spain	[59,75,83,106,107,112,118,120,123,124,125,126,127,128,129]
*Candidatus* Rickettsia barbariae	*Rhipicephalus sanguineus* s.l.	Hare, mouflon, red fox, and wolf	No data found	Cyprus, Italy, and Palestine	[70,121,130]
*Candidatus* Rickettsia goldwasserii	*Haemaphysalis adleri*, *Haemaphysalis parva*, and *Rhipicephalus sanguineus* s.l.	Golden jackal and wolf	No data found	Israel and Palestine	[113,130]
*Candidatus Rickettsia africa septentrioalis*	*Ornithodoros normandi*	Rodent	No data found	Tunisia	[131]

**Viruses**					
** *Flavivirus* **					
Louping ill	No data found	No data found	Roe deer and wild boar	France	[132]
Maeban/Maeban-like virus	*Ornithodoros maritimus*	Birds	Birds	Algeria, France, and Spain	[133]
TBEV	No data found	No data found	Rodent, roe deer, and wild boar	France and Slovenia	[132,134,135,136]
** *Orthonairovirus* **					
*Tamdy orthonairovirus*	*Hyalomma* spp. and *Hyalomma marginatum*	Rodent	No data found	Turkey	[137]
CCHF	*Hyalomma* spp., *Hyalomma aegyptium*, *Hyalomma lusitanicum*, *Hyalomma marginatum*, *Hyalomma rufipes*, and *Ixodes* spp.	Bird, red deer, and tortoise	Tortoise	Algeria, Italy, Morocco, Spain, and Turkey	[63,138,139,140,141,142,143,144]

**Table 2 microorganisms-10-01858-t002:** TBPs detected in wild animal and their ticks in the Mediterranean Rim.

Animal name	Pathogens found in ticks (^b^bacteria, ^p^parasite, ^v^virus)	Pathogens found in animals (^b^bacteria, ^p^parasite, ^v^virus)
Wild boar	*Anaplasma marginale^b^*, *A. phagocytophilum^b^*, *Babesia* spp. *^p^*, *B. occultans ^p^*, *B. rodaini ^p^*, *B. rossi ^p^*, *Coxiella burnetii^b^*, *Rickettsia* spp. *^b^*, *R. aeschlimannii^b^*, *R. africae^b^*, *R. massiliae^b^*, *R. raoultii^b^*, *R. slovaca^b^*, *Theileria* spp. *^p^*	*Anaplasma phagocytophilum^b^*, *Babesia bigemina ^p^*, *Borrelia burgdoferi* s.l. *^b^*, Louping ill, *Rickettsia slovaca^b^*, *Theileria* spp., Tick-Borne Encephalitis Virus *^v^*
Red deer	*Anaplasma marginale^b^*, *A. phagocytophilum^b^*, *Bartonella acomodis^b^*, *Ba. baciliforms^b^*, *Ba. bovis^b^*, *Ba. chomelii^b^*, *Ba. tribocorum^b^*, *Ba. vinsonii berkoffi^b^*, *Borrelia. burgdoferi* s.l. *^b^*, *Chlamydia abortus^b^*, Crimean-Congo Haemmoraghic Fever virus *^v^*, *Ehrlichia* spp. *^b^*, *Rickettsia* spp. *^b^*, *R. helvetica^b^*, *R. monacensis^b^*, *R. slovaca^b^*, *Theileria* spp. *^p^*	*Anaplasma capra^b^*, *Babesia bigemina ^p^*, *B. capreoli ^p^*, *B. divergens ^p^*, *Coxiella burnetii^b^*, *Theileria* spp. *^p^*
Bird	*A. phagocytophilum^b^*, *Babesia capreoli ^p^*, *B. venatorum ^p^*, *Borrelia afzelii^b^*, *Bo. garinii^b^*, *Bo. lusitaniae^b^*, *Bo. spielmanii^b^*, *Bo. valisiana^b^*, *Bo. turdi^b^*, *Coxiella burnetii^b^*, Crimean-Congo Haemmoraghic Fever virus *^v^*, *Ehrlichia* spp. *^b^*, Maeban virus *^v^*, *Rickettsia* spp. *^b^*, *R. aeschlimannii^b^*, *R. africae^b^*, *R. helvetica^b^*, *R. monacensis^b^*, *R. raoultii^b^*, *R. sibirica^b^*, *Theileria buffeli ^p^*, *T. orientalis ^p^*, *T. sergentis ^p^*	*Anaplasma* spp. *^b^*, Bartonella spp. *^b^*, *Borrelia burgdoferi* s.l. *^b^*, *Coxiella burnetii^b^*,*Rickettsia* spp. *^b^*
Rodent	*Anaplasma phagocytophilum^b^*, *Babesia microti ^p^*, *B. venatorum ^p^*, *Borrelia bavariensis^b^*, *Bo. burgdoferi* s.l. *^b^*, *Bo.burgdoferi* s.s. *^b^*, *Bo. crucidurae^b^*, *Bo. garinii^b^*, *Bo. hispanica^b^*, *Bo. lusitaniae^b^*, *Bo. merionesi^b^*, *Bo. valaisiana^b^*, *Coxiella burnetii^b^*, *Neoehrlichia mikurensis^b^*, *Rickettsia helvetica^b^*, *R. monacensis^b^*, *R. raoultii^b^*, *R. slovaca^b^*, *Candidatus* R. africa septentrioalis*^b^*, Tamdy orthinairovirus *^v^*, *Theileria annae ^p^*	*Anaplasma phagocytophilum^b^*, *Babesia* spp. *^p^*, *B. microti ^p^*, *Bartonella* spp. *^b^*, *Ba. elizabethae^b^*, *Borrelia* spp. *^b^*, *Bo. afzelii^b^*, *Bo. bavariensis^b^*, *Bo. burgdoferi* s.l. *^b^*, *Bo.burgdoferi* s.s. *^b^*, *Bo. garinii^b^*, *Bo. lusitaniae^b^*, *Bo. merionesi^b^*, *Bo. valaisiana^b^*, *Coxiella burnetii^b^*, *Neoehrlichia mikurensis^b^*, *Leptospira* spp. *^b^*, *L. borpetersenii^b^*, *L. interrogans^b^*, *L. kishneri^b^*, *L. santaroni^b^*, *Rickettsia slovaca^b^*, Tick-Borne Encephalitis Virus *^v^*
Wild carnivore	*Anaplasma ovis^b^*, *Chlamydia abortus^b^*, *Coxiella burnetii^b^*, *Rickettsia helvetica^b^*, *R. massiliae^b^*, *R. monacensis^b^*, *Candidatus* R. barbariae*^b^*, *Candidatus* R. goldwasserii*^b^*	*Babesia* spp. *^p^*, *B. canis ^p^*, *Cytauxzonn* spp. *^p^*, *Ehrlichia canis^b^*, *Rickettsia* spp. *^b^*, *Theileria annulata ^p^*, *T. capreoli ^p^*
Red fox	*Anaplasma ovis^b^*, *Babesia microti ^p^*, *Bartonella* spp. *^b^*, *Chlamydia abortus^b^*, *Ehrlichia canis^b^*, *Rickettsia helvetica^b^*, *R. massiliae^b^*, *R. monacensis^b^*, *Candidatus* R. barbariae*^b^*, *Theileria annae ^p^*, *T. buffeli ^p^*, *T. orientalis ^p^*, *T. sergenti ^p^*	*Anaplasma phagocytophilum^b^*, *Babesia annae ^p^*, *B. microti ^p^*, *B. vulpes ^p^*, *Coxiella burnetii^b^*, *Ehrlichia canis^b^*, *Hepatozoon canis ^p^*, *Theileria annae ^p^*
Tortoise	*Borrelia* spp. *^b^*, Crimean-Congo Hemorrhagic Fever virus *^v^*, *Rickettsia* spp. *^b^*, *R. aeschlimannii^b^*, *R. africae^b^*	Crimean-Congo Hemorrhagic Fever virus *^v^*, *Hemolivia mauritanica ^p^*
Fallow deer	*Anaplasma phagocytophilum^b^*, *Bartonella bacilliformis^b^*, *Ba. bovis^b^*, *Borrelia burdoferi* s.l. *^b^*, *Rickettsia helvetica^b^*, *R. monacensis^b^*, *R. raoultii^b^*, *R. sibirica mongolitimonae^b^*, *R. slovaca^b^*, *Theileria* spp. *^p^*	*Anaplasma phagocytophilum^b^*
Roe deer	*Babesia* spp. *^p^*, *B. microti ^p^*, *B. ovis ^p^*, *B. rodainni ^p^*, *Bartonella chomelii^b^*, *Borrelia burgdoferi* s.l. *^b^*, *Bo. burgdoferi* s.s. *^b^*, *Bo. lusitaniae^b^*, *Coxiella burnetii^b^*, *Rickettsia* spp. *^b^*, *R. aeschlimannii^b^*, *R. helvetica^b^*, *R. massilae^b^*, *R. monacensis^b^*	*Anaplasma phagocytophilum^b^*, *Borrelia burgdoferi* s.l. *^b^*, *Babesia* spp. *^p^*, *B. bigemina ^p^*, *B. capreoli ^p^*, *B. venatorum ^p^*, *Cercopithifilaria rugosicauda ^p^*, Louping ill *^v^*, *Theileria* spp. *^p^*, Tick-Borne Encephalitis Virus *^v^*
Hedgehog	*Anaplasma* spp. *^b^*, *A. ovis^b^*, *Chlamydia abortus^b^*, *Rickettsia* spp. *^b^*, *R. helvetica^b^*, *R. massiliae^b^*	
Mouflon	*Anaplasma ovis^b^*, *Chlamydia abortus^b^*, *Coxiella burnetii^b^*, *Ehrlichia canis^b^*, *Rickettsia* spp. *^b^*, *R. hoogstraalii^b^*, *Candidatus* R. barbariae*^b^*, *Theileria buffeli ^p^*, *T. orientalis ^p^*, *T. ovis ^p^*, *T. sergentis ^p^*	*Anaplasma ovis^b^*, *Coxiella burnetii^b^*, *Rickettsia* spp. *^b^*
Lizard	*Borrelia afzelii^b^*, *Bo. burgdoferi* s.l. *^b^*, *Bo garinii^b^*, *Bo. lusitaniae^b^*, *Bo. valaisiana^b^*, *Rickettsia* spp. *^b^*, *R. helvetica^b^*, *R. hoogstralii^b^*, *R. monacensis^b^*	*Hepatozoon* spp. *^p^*, *H. kisrae ^p^*
Chamois	*Anaplasma* spp. *^b^*, *Rickettsia helvetica^b^*	*Babesia capreoli ^p^*, *Theileria* spp. *^p^*, *T. ovis ^p^*
Hare	*Coxiella burnetii^b^*, *Rickettsia* spp. *^b^*, *R. massiliae^b^*, *R. monacensis^b^*, Candidatus *R. barbariae^b^*	*Babesia* spp. *^p^*, *Theileria* spp. *^p^*
Bat	*Bartonella* spp. *^b^*, *Ba. tamiae^b^*, *Coxiella burnetti^b^*, *Rickettsia* spp. *^b^*	
Porcupine		*Anaplasma phagocytophilum^b^*
Swamp deer		*Anaplasma capra^b^*, *A. phagocytophilum^b^*

**Table 3 microorganisms-10-01858-t003:** Ticks and tick-borne pathogens from wild animals on Mediterranean islands.

Country	Island	Surface Area	Western Basin/Eastern Basin	Pathogen (^A^: Positive Animals, ^T^: Positive Ticks)	Positive Ticks	Positive Ticks Host	Positive Pathogen Hosts	References
**Cyprus**	Cyprus	9251 km^2^	Eastern	*Anaplasma* spp ^A,T^., *A. ovis* ^A,T^, *C. burnetii* ^A,T^, *Rickettsia* spp. *^A^*^,*T*^, *R. aeschlimannii* ^T^, *R. hoogstraalii* ^T^, *R. massiliae* ^T^, *R. sibirica mongolotimonae* ^T^, *Candidatus* R. barbariae ^T^	*H. punctata*, *H. sulcata*, *Hy. excavatum*, *Hy. marginatum*, *Hy. rufipes*, *I. gibbosus*, *I. ventalloi*, *Rh. bursa*, *Rh. sanguineus* s.l.	Birds, mouflon, red fox	Birds, mouflon	[49,71,121]
**Greece**	Antikythira	22 km^2^	Eastern	*Rickettsia* spp. ^T^, *R. aeschlimannii* ^T^, *R. africae* ^T^	*Amblyomma* sp., *Haemaphysalis* sp., *Hy. marginatum*, *Hy. rufipes*, *Ixodes* sp., *I. frontalis*, *Rhipicephalus* sp.	Birds	No data found	[108]
**France**	Corsica	8722 km^2^	Western	*Leptospira borpetersenii* ^A^, *L. interrogans*^A^, *R. aechlimannii*^T^, *R. massiliae*^T^, *R. monacensis*^T^, *R. slovaca*^T^, *Candidatus* R. barbariae ^T^	*D. marginatus*, *Hy. aegyptium*, *Hy. marginatum*, *Hy. rufipes*, *I. ricinus*, *Rh. bursa*	Birds, tortoise, and wild boar	Rodent	[102,112,114]
**Italy**	Asinara	51.90 km^2^	Western	CCHF ^T^	*Haemaphysalis* spp., *H. punctata Hyalomma* sp., *I. ricinus*	Birds	No data found	[183]
	Capri	10.40 km^2^	Western	*A. phagocytophilum*^T^, CCHF ^T^, *Rickettsia* spp. ^T^, *R. aechlimannii* ^T^, *R. africae* ^T^	*Amblyomma* sp., *Haemaphysalis* sp., *H. punctata*, *Hyalomma* sp., Hy. *marginatum*, *Hy. rufipes*, *Ixodes* sp., *I. frontalis*, *I. ricinus*, *Rhipicephalus* sp.	Birds	No data found	[60,108]
	Pianosa	10.25 km^2^	Western	*Bo. spielmanii*^T^, *Bo. valaisiana*^T^, CCHF ^T^, *R. aeschlimannii*^T^, *R. helvetica*^T^	*Haemaphysalis* spp., *H. punctata*, *Hyalomma* spp., *Hy. dentritum*, *Hy. marginatum*, *I. accuminatus*, *I. ricinus*, *I. ventalloi*	Birds	No data found	[92,183]
	Ponza	7.300 km^2^	Western	CCHF ^T^, *R. aeschlimannii* ^T^, *R. africae* ^T^, *R. raoultii* ^T^	*A. marmoreum*, *Haemaphysalis* spp., *H. punctata*, *Hyalomma* spp., *Hy. marginatum*, *I. frontalis*, *I. ricinus*, *I. ventalloi*	Birds	No data found	[110,183]
	Sardinia	24,090 km^2^	Western	*A. ovis*^T^, *A. phagocytophilum*^T^, *B. bigemina*^T^, *Bartonella* spp. ^T^, *C. burnetii* ^T^, *Ch. abortus* ^T^, *Ch. psitacci* ^T^, *E. canis* ^T^, *Rickettsia* spp. ^T^, *R. aeschlimannii* ^T^, *R. conorii israelensis* ^T^, *R. helvetica* ^T^, *R. hoogstraalii* ^T^, *R. massiliae* ^T^, *R. slovaca* ^T^, *Candidatus* R. barbariae ^T^, *T. buffeli/sergneti/orientalis* ^T^, *T. equi* ^T^, *T. ovis* ^T^	*D. marginatus*, *H. punctata*, *H. sulcata*, *Hy. marginatum*, *I. festai*, *Rh. annulatus*, *Rh. bursa*, *Rh. pusillus*, *Rh sanguineus* s.l.	Birds, hedgehog, marten, mouflon, red deer, red fox, wild boar	No data found	[46,70,93,101,120,124,126]
	Sicily	25,711 km^2^	Western	*A. marginale*^A^, *A. phagocytophilum*^A^, *A. ovis*^A^	No data found	No data found	Porcupine, rodent	[69]
	Ustica	8.240 km^2^	Western	CCHF ^T^	*Haemaphysalis* spp., *H. punctata Hyalomma* spp., *I. ricinus*	Birds	No data found	[183]
	Ventotene	1.540 km^2^	Western	CCHF ^T^	*Hy. rufipes*	Birds	No data found	[143]
	Zannone	0.9 km^2^	Western	CCHF ^T^	*Haemaphysalis* spp., *H. punctata Hyalomma* spp., *I. ricinus*	Birds	No data found	[183]
**Spain**	Columbretes	0.19 km^2^	Western	Maedan virus ^A,T^	*O. maritimus*	Birds	Birds	[133]
	Isla Grosa	17.5 km^2^	Western	Maedan virus ^A,T^	*O. maritimus*	Birds	Birds	[133]
	Mallorca	3640 km^2^	Western	Maedan virus ^A,T^	*O. maritimus*	Birds	Birds	[133]
	Medes	0.215 km^2^	Western	Maedan virus ^A,T^	*O. maritimus*	Birds	Birds	[133]

## Data Availability

The authors confirm that the data supporting the findings of this study are available within the article and its supplementary material. Raw data that support the findings of this study are available from the corresponding author, upon reasonable request.

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
