# Peer review of "A Systematic Review of the Distribution of Tick-Borne Pathogens in Wild Animals and Their Ticks in the Mediterranean Rim between 2000 and 2021"

_microorganisms, 2022, doi:10.3390/microorganisms10091858_

Round 1
Reviewer 1 Report
Defaye et al submitted a very interesting manuscript exhaustingly describing ticks and tick-borne pathogens (TBPs) of wild-living animal in the Mediterranean Rim. Despite their work is admirable in terms of the effort and the number of publications included and despite similar publication is needed and can serve as a very important base stone for numerous other research projects, I cannot recommend the manuscript for publication unless major changes and improvements will be done.
1) Despite the work exhaustively describes the ticks and TBPs found on wild-living animals it totally omits the wider contexts and it does not discuss the findings in context of ticks and TBPs found on the same locality but collected from vegetation, livestock, companion animals, or humans. Wild-living animals are not a closed system and therefore the fact that some ticks are frequently found on free farmed livestock on a particular locality but not on the wild-living animals does not mean that it does not feed on them. Similarly, what can be a host of a tick collected from a vegetation in the middle of a national park far from any pasture?
2) The authors should realize and stress it through all the manuscript that their data are extremely biased in many aspects and because of many reasons. May be the most visible bias is in research intensity between various regions (102 publications from Western Europe versus 10 publications from North Africa). It shows us that the data from North Africa should be handled with high wariness and discussed in respect to similar research done on ticks and TBPs collected from other hosts. Similarly, usage samples from different hosts is also biased by availability of the samples, difficulty to get legal permits for sampling, local research habits (e.g., existence of tradition in research focused on a particular tick, TBP, or host species in a particular country etc.), or just by chance. There is almost nothing which authors can do with this kind of bias, but they should be aware of it and stress it out as much as possible.
3) The related issue is that the authors focus just on the information, which are present in literature, and they do not stress out the information, which are lacking, which is, according to my opinion, one of the most important qualities of a good systematic review. For example, in the case of TBPs the authors stress out the countries where the pathogens were found but they do not mention the countries where they were not found despite the relevant effort was put in the search for them (a relevant number of relevant samples was screened). There is a big difference if somebody proves that the pathogens/tick is missing on a particular locality compared to the fact that nobody just tried to look for them there. This kind of information would dramatically increase the value of the review.
4) The way how the geographical locality is defined is a little bit problematic. I understand that to use the whole countries is the easiest choice, but we must realize that large countries such as France, Turkey, or Algeria are not located only in the Mediterranean region and for example in case of France there would be a great difference between Corsica or the French Riviera on one hand and Brittany or Alsace on the other. Therefore at least in case of France, Algeria, and Turkey the finest division would be useful or at least it could be great to show from exactly which parts of these countries the data came.
5) During the preparation of the data for this manuscript, the authors had to collect and extract enormous amount data from the selected literature preparing the excel table as described in the methods. It would be great if they could provide this raw data as an supplementary material to this review as well.
I totally understand that these requirements are very demanding on the authors and that their inclusion would lead to increase of size of the submitted manuscript, but I think that it would also lead to dramatic increase of its informative value.
Apart these major comments I also have several minor issues which should be improved but they may be totally leaved out from the manuscript during the major changes and therefore they has not be relevant in the next round of review process:
Figure 2 - 2021 cannot be showed on the figure as only first two months were screened.
Lines 160-163 - Use either bullets or numbers to separate the three reasons.
Line 173 - Cultivation of detected TBPs was not used in any of selected papers?
Line 183 - an extra dot
Line 384 - the reference is not in correct format
Chapter 4 - It is not apparent from the text if the mentioned prevalence is for the pathogens themselves (detected either directly by some microscopic techniques or indirectly by detection of their nucleic acids in the samples) or if it refers to the seroprevalence of antibodies detected in sera of potential animal hosts.
Chapter 4 – I would suggest mentioning ticks first and then pathogens as presence of tick vectors influence which tick-borne pathogens can be expected.
Figure 4 - The color codes are hard to read, the used colors are too similar to be easily distinguishable at least at the computer screen.
Figure 4a - It would be great to show the dots on the exact localities where the samples described in selected publications were collected.
Despite it seems that I have many comments and I rise many issues I would like to stress that I think that such kind of manuscript deserves to be published and after incorporation of my comments into the manuscript it would be a pleasure for me to recommend it for publication.
PS: Are there really no publications about ticks and TBPs of wildlife from Crete, Greek islands in Aegean Sea or Dalmatia islands? If not, they can be a good tip for a next field trip... ;-)
Author Response
Dear reviewer,
We thank you for all your comments. You will found our answer and corrections in the attached file.
Kind regards

Reviewer 2 Report
This systematic review concerns the distribution of tick-borne pathogens, their vectors and reservoirs in the Mediterranean based on papers published between 2000 and 2021. Overall, the work is well structured and contains useful information; however, in my opinion, the content of the paper seems to be more suitable for the Pathogens journal.
The literature review was performed correctly, the data obtained merit publication. However, the paper contains several deficiencies that should be corrected before publication. I also have some suggestions for tables and figures.
1. Material and Methods - This chapter is too elaborate; I suspect that the audience of this review will not be interested in the methodology of searching the database. Figure 2 makes sense if it were to show readers that microscopic analyses have been fully supplanted from this type of research.
2. The paragraph marks in the lines L159-163 should be removed.
3. The format of section titles should be standardized, e.g., 4.1 “Parasites” in italics without bold and 4.2 “Bacteria” in bold simple font or giving only the generic name in the title or adding "genus" next to ticks, etc.
4. Each sections should be described in the same way, i.e. starting with a taxonomic description of a specific TBP, then it should be clearly distinguished whether general information is given, from different geographic zones, or those specific to the Mediterranean area. For example, L198-199 it is not clear whether the information " The zoonotic species are principally transmitted by ticks belonging to the genus Ixodes" refers to the results of this work, or is general information on zoonotic Babesia. If it is the latter, it is untrue, because zoonotic B. microti is transmitted in Europe also by Dermacentor (e.g., https://doi.org/10.1186/s13071-021-05019-3) or Rhipicephalus, not only by Ixodes.
5. Table 1, which contains the most important results for the reader, should be divided into three separate tables for parasites, bacteria and viruses. Such a presentation of the results will be more convenient to view. (Although in fact all TBPs are parasites, so this division is practical, not scientific).
6. Check text in the tables carefully – e.g. martes or martens?
7. Page 12: Each apicomplexan (as each species at all) belongs to a family and you do not need to repeat that it is apicomplexan because it is in the title of the section. Of course some systematic information could be delivered here, e.g., the specific family to which it belongs, as shown in the case of bacteria. This is another example of how the text is not coherently edited.
8. Information in lines 230, 238-240, 273-274, 403-405, 426-428, 463-464, 496-497 – all need references.
9. Lines 251-254 - The authors confuse the term zoonotic; it does not mean "infecting animals," but infecting humans through animals.
10. L304 – Currently there is only one genus Chlamydia (e.g., doi:10.1093/gbe/evv201), therefore only this name should be in the section title and in the table, while in the text you can write that what was noted in the publications concerned the morphotype once considered the genus Chlamydophila (because this one is listed in the table 1!).
11. The text in the section describing viruses should also be written in a uniform way; now, once the systematics is given (e.g., Flaviviruses, at least at the group level), and other times this information is absent (Orthonairovirus).
12. The term animals is used vaguely in some cases because ticks are animals too; it is better when wild or domestic is added.
13. Line 415 – Remove “Genus” in titles here and below.
14. Line 441 – References are provided in wrong format.
15. Fig. 4 is too small and thus unreadable; maybe Figures 4A and 4B can be omitted, because this information is in the main text (or transferred there to the description), and 4C and 4D should be left in a vertical layout, where they would be larger and more legible.
16. Overall analysis of the four areas of the Mediterranean Rim is quite artificial. Countries classified as Western Europe are in fact Southwestern Europe. Besides, apart from the border, there is probably nothing separating northern Italy from some of the Balkan countries. In general, this part of the review is the weakest substantively, because it does not refer to physical geography (e.g., marine regions), but to political one (countries).
17. Table 2 is wrongly designed and unclear. Information about numbers of countries is an artifact of the intensity of research in the country, and has little to do with the actual prevalence of TBP.
18. Table S2 is much better arranged and would be more useful to readers; should be in the body text.
19. Remove first two sentences and start the next “Western Europe-group countries”.
20. Review concerning analysis of the four areas of the Mediterranean Rim should focus on organisms (TBP, ticks and reservoirs) instead of paper numbers.
21. Line 685 – where have been these hosts reported?
22. Conclusions contain results, while they should focus on general statements, e.g., should focus on the fact that most areas of the Mediterranean Rim remain unexplored, that ranges of TBP or ticks are changing, how biologically different the geographic regions being compared are, etc.
Author Response
Dear Reviewer,
We thank you for all of your comment. You will found our comments and corrections in the attached files.
Kind regards,
